# Fabrication and Characterization of Bio-Epoxy Eggshell Composites

**Stephen Owuamanam [1], Majid Soleimani [2] and Duncan E. Cree [1,*]**

[1]   Department of Mechanical Engineering, University of Saskatchewan, Saskatoon, SK S7N 5A9, Canada; stephen.owuamanam@usask.ca
[2]   Department of Chemical and Biological Engineering, University of Saskatchewan, Saskatoon, SK S7N 5A9, Canada; mas233@mail.usask.ca
[*]   Correspondence: duncan.cree@usask.ca; Tel.: +1-306-966-3244

**Abstract:** In this study, an innovative composite was fabricated in which the matrix is partially derived from natural sources and the filler from undervalued eggshell waste material. The effect of coating eggshells and mineral limestone with 2 wt.% stearic acid on the mechanical properties of a bio-epoxy matrix was investigated. Eggshells and limestone (untreated and stearic acid-treated) fillers were added to the bio-epoxy matrix in quantities of 5, 10, and 20 wt.% loadings using a solution mixing technique. The $CaCO_3$ content in eggshells was confirmed to be 88 wt.%, and the crystalline phase was found to be calcite. The stearic acid coating did not show any decrease in crystallinity of the fillers. Scanning electron microscopy (SEM) displayed changes in the fractured surfaces, which infers the fillers altered the bio-epoxy polymer. The mechanical property results showed enhancements in the composite tensile modulus and flexural modulus compared to the pure bio-epoxy, as expected. In contrast, despite the improvement in the tensile and flexural strengths of the stearic acid-treated fillers, the composite strength values were not higher than those of the unfilled bio-epoxy matrix. The energy absorbed by all composites in Charpy impact tests fell below that of the pure bio-epoxy and decreased with an increase in filler content for both untreated and stearic acid-treated fillers tested at 23 and −40 °C. Statistical analysis of the results was conducted using Statistical Analysis Software (SAS) with ranking based on Tukey's method. The study identified that the addition of 5, 10, and 20 wt.% in a bio-epoxy matrix may be acceptable provided the end product requires lower tensile and flexural load requirements than those of the pure bio-epoxy. However, filler loadings below 5 wt.% would be a better choice.

**Keywords:** eggshell waste; calcium carbonate; bio-epoxy; composite; mechanical testing

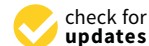



## 1. Introduction

In recent years, there has been an uptake in the use of polymer composites for various applications due to their light weight, high specific stiffness, and moderate strength properties. During and after World War II (1940s) there was an increase in the production and development of synthetic polymers and plastics produced from petroleum resources [1]. Synthetic epoxy is frequently utilized in diverse applications due to its simple processing methods, good elevated temperature properties, and exceptional strength. Generally, epoxy polymers have a higher purchase cost than other thermoset polymers; however, in the aerospace industry, more than two-thirds of polymers utilized are based on epoxy [2]. As a green alternative, bio-based epoxy resins are synthesized from modified plant oils, sugars, polyphenols, and terpenes [3], where resins contain 31–55% bio-contents. Therefore, bio-based epoxy resins could serve as substitutes to petroleum-based epoxy resins.

Because the cost of polymers can be relatively high, it has been customary to add lower cost fillers to reduce the overall cost of the raw composite materials while in some cases improving their properties. Limestone, a mineral with a high calcium carbonate content ($CaCO_3$) is a common filler used to replace part of the more expensive polymer matrix and can also improve the stiffness of the components. Another source of calcium carbonate

not widely discussed is waste chicken eggshells. Recent studies revealed eggshells contain 94–96 wt.% $CaCO_3$ in the form of calcite, 3–4 wt.% organic matter, and negligible traces of magnesium, phosphorus, and other elements [4–6].

Egg breaking plants (sometimes referred to as "breaker plants") produce enormous quantities of eggshell waste as a by-product and are largely discarded in landfills. Eggshells are compostable because they can decompose and add nutrients to the soil. However, issues arise when the quantity of discarded eggshells is very large, i.e., when measured in tons. During their decomposition, ammonia ($NH_3$) and hydrogen sulfide ($H_2S$) are emitted, which can be unfavorable for the environment, and the produced salmonella bacteria can be harmful to humans. More importantly, the cost of landfilling waste eggshell is significant. For instance, breaker plants in Europe and in the United States of America spend approximately USD 100,000 annually to dispose of eggshells in landfills [7]. One means to redirect waste eggshells is to use them in polymers as fillers. A recent review article estimated the amount of $CaCO_3$ that could be recovered from waste eggshells in the top ten annual egg producing countries [8]. For example, China's annual production of eggs in 2017 was 536,818,007,000. Of this amount, 30% of eggs are generally sent to breaker plants, which amounts to 161,045,402,000 eggs. Based on the dry empty shell weighing 6.6 g, about 1,062,899,654 kg of $CaCO_3$ could be recovered. In practical terms, ~35,429,988 bags of $CaCO_3$ each weighing 30 kg can be recovered from eggshell waste annually for use in various applications. Repurposing waste eggshells can be seen as a cost-effective alternative to paying disposal tipping fees at the landfill.

When $CaCO_3$ filler particulates are incorporated into polymers, agglomeration has been recognized as a disadvantage in several studies [9–11]. The cluster of fine particles was reported to cause micro-cracks, which are alleged to lead to brittle fractures in polymer composites [12]. In an effort to enhance dispersion of fillers and improve bonding at the interface between matrix and filler, the most widely used surface coating on $CaCO_3$ powder has generally been stearic acid, an inexpensive fatty acid. The quantity of stearic acid required to effectively coat $CaCO_3$ powders has been investigated [9,13]. In one study, 0.5–4 wt.% of stearic acid was coated on the surface of calcite fillers. The amount necessary to coat the surface of the calcite particles with a single layer of stearic acid was suggested to be between 1.5 and 2 wt.% [13]. In a recent study, eggshell powder was modified with 1–3 wt.% stearic acid and added to a synthetic epoxy resin. The tensile elongation property showed an improvement when 2.5 wt.% stearic acid was used compared to the pure epoxy, which the authors attributed to hydrogen bonding between stearic acid coated particles and the matrix [9]. The literature suggests that a lower amount of stearic acid is more effective.

This work reports the fabrication of bio-epoxy composites containing untreated and stearic acid-treated $CaCO_3$ powders, derived from waste eggshells and conventional mineral limestone. The tensile strength/modulus, flexural strength/modulus, and Charpy impact energy of bio-epoxy composites containing different amounts of $CaCO_3$ fillers were compared against those of bio-epoxy. Inductively coupled plasma mass spectrometry (ICP-MS) was used to determine the $CaCO_3$ content in the prepared eggshell filler. X-ray diffraction (XRD) identified the crystalline phase of both eggshell and limestone powders. Scanning electron microscopy (SEM) was used to view the $CaCO_3$ filler morphologies and fractured surfaces of the composites. Differential scanning calorimetry (DSC) was implemented to assess whether the fillers had an effect on the glass transition temperature ($T_g$) of the composite materials.

## 2. Materials and Methods

### 2.1. Materials

The matrix was a green bio-epoxy polymer (Super Sap CPM resin and CPL hardener) obtained from Entropy Resins Inc., San Antonio, TX, USA, and containing a bio-based content of 31%. The mix ratio of resin: hardener (by weight) was 100:40. The fillers were white eggshell obtained from eggshell waste and mineral limestone was purchased from Imasco Minerals Inc., Surrey, BC, Canada. Silicone molds for specimen fabrication were

made from Mold Max® 10T (Smooth-On Inc., Macungie, PA, USA). Stearic acid in the form of flakes was acquired from Acros Organics, 97% purity (Fisher Scientific., Ottawa, ON, Canada).

### 2.2. Eggshell Powder Preparation

The as-received waste eggshells were prepared by coarse crushing, rinsing with water, and drying at 105 °C for 24 h to eliminate water, odor, and contaminants from the organic membranes and egg remnants [4]. The coarse ground eggshells were further ball milled for 6 h into fine powders using a small-scale milling apparatus to further reduce the particle size. Distilled water was added to the fine eggshell powders and manually agitated, and a precipitation process was used to remove additional membranes followed by a drying step. After a vigorous agitation, sufficient time was given to allow the denser eggshell particle suspension to settle to the bottom of the pail, while the lighter eggshell membranes floated to the top. By slowly tipping the container, the membranes flowed out of the pail. The eggshell and limestone powders were each sieved to 32 μm (No. 450) mesh using a vibrating sieve shaker machine (Ro-tap).

The eggshell and limestone fillers were treated with 2 wt.% stearic acid as proposed in a previous study [13]. Individually, 100 g of eggshell or limestone powder was added to a beaker containing 100 mL of ethanol and 300 mL of distilled water at a weight/volume/volume ratio (1:1:3) (powder: ethanol: water). The solution was stirred using a magnetic stirrer for 2 h at room temperature (23 °C) in order to completely wet the particles. In a separate glass beaker, to obtain a 2 wt.% stearic acid in the final solution, 9.156 g of stearic acid chips was dissolved in 100 mL (78.8 g) of ethanol. The powder/ethanol/distilled water mixture was stirred and heated until it reached 80 °C and the 2 wt.% stearic acid solution was added dropwise over a 30 min period. The mixture was stirred for an additional 2.5 h to ensure homogenization and finally filtered using a filter paper of particle retention 5–10 μm. The stearic acid coated powder was subsequently dried at 105 °C for 24 h and agglomerated particles were separated using a mortar and pestle.

### 2.3. Particle Size Analysis

Particle size analysis was undertaken with a Malvern Mastersizer 2000 S (long bench) laser diffraction particle size analyzer using a dry dispersion method. The apparatus uses two laser light sources consisting of a blue light (wavelength 466 nm) and a red light (wavelength 633 nm) to obtain the particle size distribution. The process was repeated three times to ensure reproducibility and the average results are reported.

### 2.4. Bio-Epoxy/Calcium Carbonate Composite Preparation

Four fillers (untreated eggshell, untreated limestone, eggshell treated with stearic acid, and limestone treated with stearic acid) were added in separate batches to the bio-epoxy matrix using a solution mixing method. Formulations consisted of adding three different weight fractions of the fillers as replacements of the matrix in loadings of 5, 10, and 20 wt.%, and are listed in Table 1. Prior to dispersing the fillers in the matrix, the eggshell and limestone powders were degassed (shell lab Model SVAC1E) under vacuum >28 inHg for 30 min at 23 °C to eliminate air in the pores of the particulates. The required amount of bio-epoxy resin was first poured into a 2000 mL polypropylene beaker followed by the addition of the appropriate amount of filler. The mixture was dispersed using a magnetic stirrer rotating at 700 rpm for 1 h. To improve dispersion of the particles within the bio-epoxy, an ultrasonic homogenizer (Model FS-900N, MXBAOHENG, Shanghai Shengxi Co. Ltd., Shanghai, China) was used to further disperse the fillers for 5 min. In Bittmann et al. [14], a duration of 5 min. was suggested for dispersion of fillers using an ultrasonic homogenizer to avoid degradation of the epoxy mechanical properties. Excessive sonication may increase the temperature and viscosity, which could prevent effective dispersion of fillers due to the start of resin curing without addition of the hardener [15]. The liquid resin/filler mixtures were degassed under vacuum (28 inHg) for 30 min to evacuate air bubbles generated from

mixing. Subsequently, the necessary amount of hardener was added and stirred manually for 3 min followed by degassing for 15 min. The bio-epoxy formulations were slowly poured into silicone rubber molds having specific geometries for specimen testing. The Mold Max 10T was mixed at a weight ratio of 100:10 (silicone rubber: hardener) and cured for 24 h at room temperature and demolded. The silicone rubber molds were subjected to a post-cure at 65 °C for 4 h as instructed by the supplier's data sheet to improve the tearing properties. The composites were cured for 24 h at room temperature and post-cured at 82 °C for 40 min [16].

**Table 1.** Formulations of filler in the bio-epoxy composites.

| Filler (wt.%) | Resin (wt.%) | Hardener (wt.%) |
|---|---|---|
| 0 | 71.43 | 28.57 |
| 5 | 67.86 | 27.14 |
| 10 | 64.29 | 25.71 |
| 20 | 57.14 | 22.86 |

*2.5. Measurements and Characterization*

ICP-MS was used to determine the calcite or $CaCO_3$ content in the prepared eggshell filler. The powdered samples were digested in 65 wt.% concentrated nitric acid at 200 °C in a closed vessel employing a microwave digestion system (Model CEM Corporation Mars 6, Matthews, NC, USA) using 1800 watts of microwave energy at a frequency of 2450 MHz. An ICP-MS (Agilent 7700x ICP-MS, Tokyo, Japan) was used to measure elemental composition. The tests were performed in triplicate and averaged. The obtained $CaCO_3$ composition of eggshells was compared with mineral limestone as reported in the supplier's technical data sheet. XRD was used to identify the crystalline phase of both eggshell and limestone powders using an X-ray diffractometer (Ultima IV, Rigaku Americas Corporation, Woodlands, TX, USA) with Cu Kα radiation operating at 40 kV and 44 mA, at a scanning rate of 4.0°/min and a diffraction angle (2θ) between 20° and 50°. Filler powder morphology, and tensile and flexure fractured surfaces, were observed by a scanning electron microscope (SEM) (JEOL JSM-6010 LV) (Tokyo, Japan) operated at 7–15 kV. The samples were sputter gold coated to make them conductive before being examined. Tensile and three-point flexural tests were conducted using an Instron universal testing machine (Model No. 3366, Instron Corp., Norwood, MA, USA) equipped with a 5 kN load cell. Tensile strength was evaluated at a cross-head speed of 10 mm/min according to ASTM D638-14 [17]. The dog-bone sample dimensions were 165 × 13 × 3.2 mm (l × w × t). Flexural sample dimensions were 200 × 20 × 10 mm (l × w × t) with a span to depth ratio of 16:1 as recommended by the ASTM D790-17 standard [18]. The cross-head speed was 4 mm/min. Charpy impact tests were performed with an Instron Model 450 MPX (Instron, Norwood, MA, USA) series impact tester. In this test, ASTM standard D6110-18 [19] was used as a guide to compare the impact toughness of the composites at various filler loadings with respect to the unfilled bio-epoxy matrix. The tests were performed on un-notched, non-standard specimens of dimensions 55 × 13 × 3.2 mm (l × w × t). It was reported that un-notched samples were better for determining the presence of agglomerates from powder fillers added to composite materials compared to notched test specimens that are insensitive to agglomerates [20]. To determine if temperature had an influence on the impact behavior of the composites, the samples were conditioned at 23 °C and −40 °C for 4 h in a Tenney Environmental Chamber (Tenney Engineering Inc., Parsippany, NJ, USA) prior to testing. Tensile, flexural, and Charpy impact tests utilized an average of five specimens for each composite formulation. The $T_g$ of the composite materials was investigated using a DSC (Model 2910 V4.4E, TA Instruments, New Castle, DE, USA) according to the ASTM E1356-08 (2014) [21] standard. Samples were prepared in a powder form of approximately 7–10 mg and placed in an aluminum pan and heated from 22 to 80 °C at a heating rate of 10 °C min$^{-1}$ under nitrogen atmosphere. Statistical analysis of

the data was conducted using Statistical Analysis Software, SAS 9.4 (SAS Institute, Cary, NC, USA) with ranking based on Tukey's method.

### 3. Results and Discussion

*3.1. Chemical Composition Analysis (Powders)*

ICP-MS elemental analysis of eggshells revealed a comparatively high $CaCO_3$ content of 88 wt.% $\pm$ 0.71 with traces of magnesium (0.27%) and phosphorus (0.13%), compared to 99.9 wt.% $CaCO_3$ content for limestone as reported in the supplier's data sheet. The variations in $CaCO_3$ chemical composition contents from the literature may be due to the different sample preparation for different chemical analysis techniques. In addition, in an earlier study on the same waste eggshell batch, inductively coupled plasma-optical emission (ICP-OES) was performed and the results showed elements of chloride (0.058%), sulphate (0.034%), and sulfur (0.067%) [4]. The minor difference in composition was anticipated because the eggshells contain organic matter which is absent in limestone. Although significant attempts were made to remove organic membranes using an agitation technique, they were still present in the samples. A recent article showed eggshells have an inner and outer membrane where the inner membrane is easier to remove than the outer membrane. The study suggested heating eggshells to 450 °C would allow the organic membranes to be removed without changing the composition of $CaCO_3$; however, there is an apparent color change to gray with carbon remnants [22]. It is well known that calcination of $CaCO_3$ above 800 °C causes conversion into calcium oxide (CaO). A slightly higher heat treatment below the formation of CaO may remove the carbon. For composite materials, depending on the color or type of polymer matrix, color change may not have an impact on the end product. In another study, a 10% bleach treatment soaked for 48 h or a 50% bleach treatment soaked for 10 min was able to remove the organic membrane to produce pure calcite [23]. The aim of this current work was to avoid the use of chemicals and/or additional processing.

*3.2. X-ray Diffraction Analysis (Powders)*

XRD patterns were obtained for untreated eggshells, untreated limestone, eggshell treated with stearic acid, and limestone treated with stearic acid, in addition to stearic acid particulates for comparison, and are presented in Figure 1. The diffraction peaks of eggshell and limestone suggested the crystalline phase to be $CaCO_3$ in the form of calcite. As previously reported, the most thermodynamically stable polymorph of $CaCO_3$ is calcite [24]. The major intensity peak was found at a 2θ angle of 29.4°, while minor peaks occurred at 23.2°, 31.5°, 36.1°, 39.6°, 43.3°, 47.7°, and 48.7° for both eggshell and limestone, as also reported in the literature [25]. Stearic acid presented two characteristic peaks at 21.5° and 24.1°. The diffracted peaks of stearic acid particulates did not show on the eggshell, limestone, or both stearic acid-treated fillers. This suggests that the stearic acid content (2 wt.%) used to treat the fillers was too small to be identified by the XRD detector. In a previous study, XRD diffraction peaks of 2.5 wt.% stearic acid were not apparent on stearic acid coated calcium carbonate particles. However, the presence of stearic acid is expected to reduce the diffraction peak intensities of calcium carbonate without changing the crystalline structure [9]. Stearic acid has a low crystallinity and tends to decrease the diffraction peak intensities.

*3.3. Scanning Electron Microscopy Analysis (Powders)*

The micrographs of eggshell, limestone, eggshell treated with stearic acid, and limestone treated with stearic acid fillers are presented in Figure 2a–d. Both untreated and treated fillers have similar structures showing coarse and irregular morphologies, possibly due to both filler types being reduced by a crushing/grinding process. In general, the eggshell (Figure 2a) and limestone (Figure 2c) tend to exhibit a rhombohedral-like morphology, suggesting the existence of calcite crystals [5]. The eggshell particles treated with

stearic acid (Figure 2b) and limestone particles treated with stearic acid (Figure 2d) had fewer rough/sharp edges, possibly due to the applied coatings.

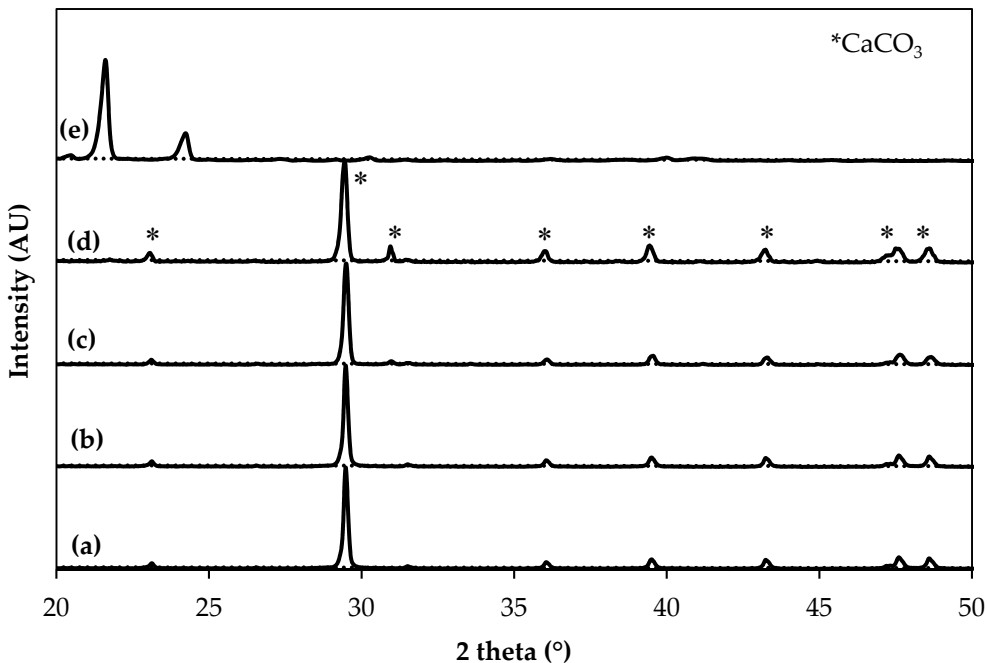

**Figure 1.** X-ray diffraction (XRD) patterns of (**a**) eggshells, (**b**) eggshells treated with stearic acid, (**c**) limestone, (**d**) limestone treated with stearic acid, and (**e**) stearic acid particulates.

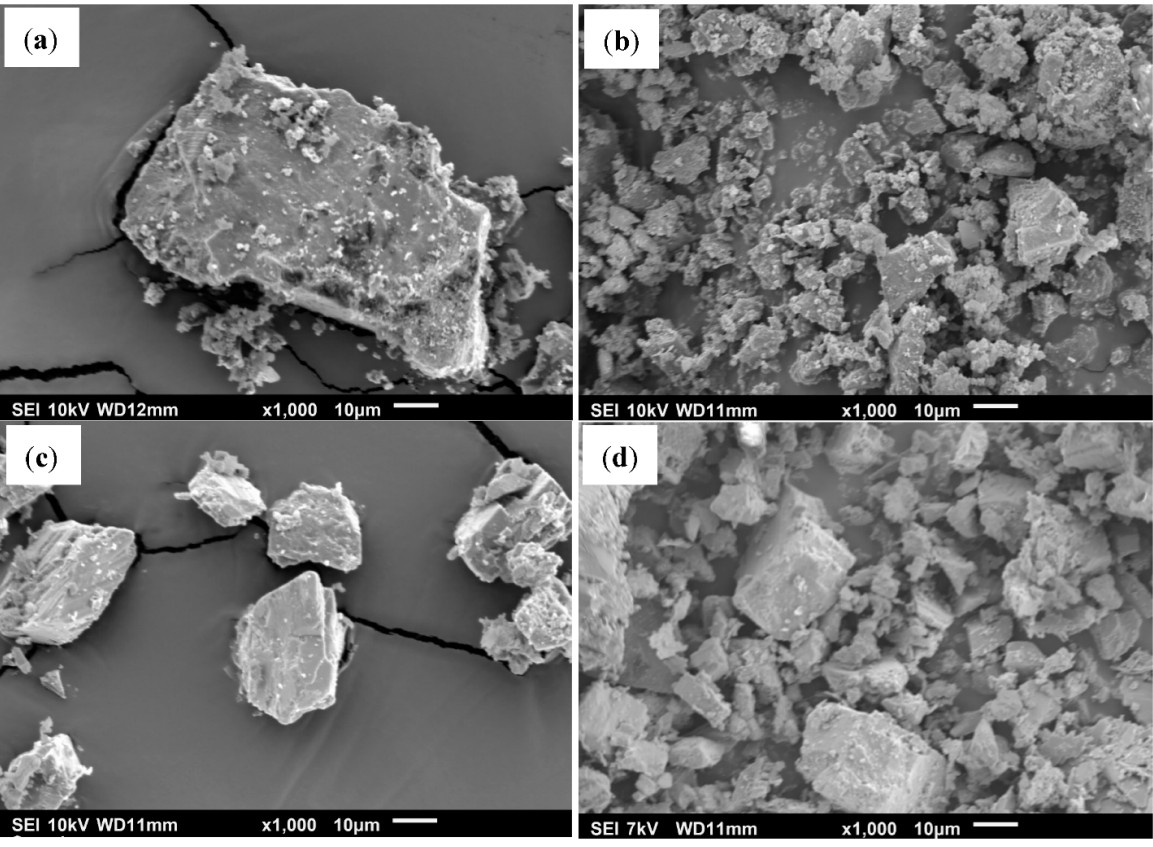

**Figure 2.** SEM micrographs showing particle morphologies of (**a**) eggshell, (**b**) eggshell/stearic acid treated, (**c**) limestone, and (**d**) limestone/stearic acid treated.

### 3.4. Particle Size Analysis

The particle size distribution curve (PSDC) of the prepared eggshell (ES), eggshell treated with stearic acid (ES/SA), limestone (LS), and limestone treated with stearic acid (LS/SA) are shown in Figure 3. The average particle size based on volume distribution was determined to be 21.2 ± 2.0 µm, 11.5 ± 1.0 µm, 25.1 ± 2.2 µm and 12.8 ± 2.2 µm, respectively. Slight differences in untreated eggshell and limestone powders may be attributed to the particle reduction method. As shown in Figure 3a,b, the eggshell and limestone fillers had a broader particle size distribution range compared to the stearic acid treated fillers, which had a narrower range. Comparable results in particle size distribution were obtained for 1 wt.% stearic acid-coated $CaCO_3$ particles [26]. The differences were believed to be due to larger agglomerated particles being easily separated into smaller lumps due to the stearic acid coating.

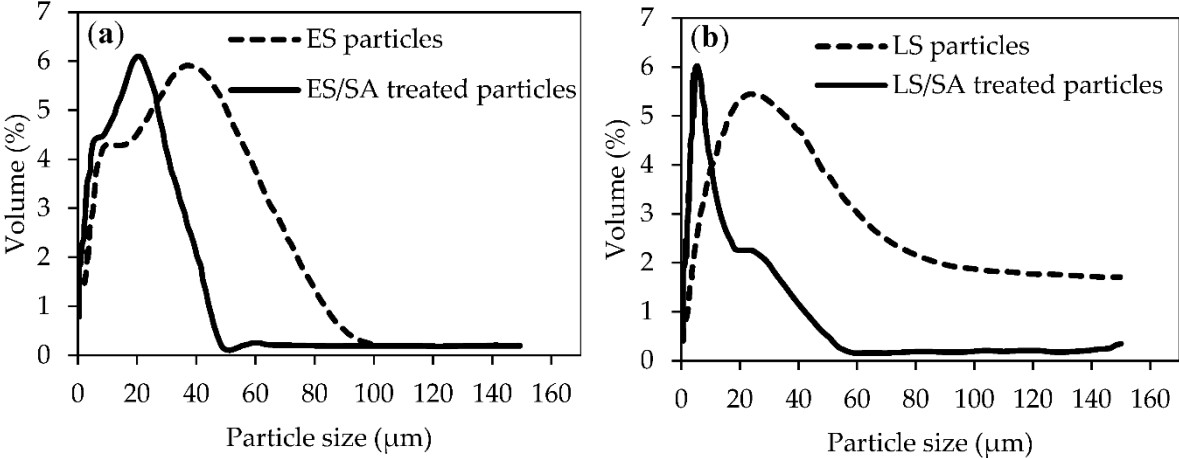

**Figure 3.** Particle size volume distribution curves for (**a**) eggshell (ES) and eggshell/stearic acid treated (ES/SA) and (**b**) limestone (LS) and limestone/stearic acid treated (LS/SA).

### 3.5. Scanning Electron Microscopy Analysis (Fractured Surfaces)

Figure 4 shows the tensile fractured surfaces of the unfilled bio-epoxy (Figure 4a), bio-epoxy containing 5 wt.% untreated eggshells (Figure 4b), bio-epoxy containing 5 wt.% eggshells treated with stearic acid (Figure 4c), bio-epoxy containing 20 wt.% untreated eggshells (Figure 4d), bio-epoxy containing 20 wt.% eggshells treated with stearic acid (Figure 4e), bio-epoxy containing 5 wt.% untreated limestone (Figure 4f), bio-epoxy containing 5 wt.% limestone treated with stearic acid (Figure 4g), bio-epoxy containing 20 wt.% untreated limestone (Figure 4h), and bio-epoxy containing 20 wt.% limestone treated with stearic acid (Figure 4i). The loadings of 5 and 20 wt.% were selected to view morphologies of the composites containing minimum and maximum filler contents. A similar approach was followed where SEM fractured surfaces were reported for only 10 and 30 wt.% for a study that added 0, 10, 20, 30, and 40 wt.% eggshell fillers in a polypropylene matrix [10]. The fractured surface of the unfilled bio-epoxy resin had smooth and cleavage features indicative of a brittle fracture. It also showed fewer uninterrupted crack paths after initiation compared to the composites. Similar fractured surfaces were observed with untreated and stearic acid treated composites. The micrographs of the fractured surfaces for composites containing 5 wt.% fillers (Figure 4b,c,f,g) showed a greater degree of roughness (white ridges), which may be due to more crack initiation and propagation sites as a result of filler additions. The degree of roughness further increased at 20 wt.% filler loadings (Figure 3h,i and Figure 4d,e). Overall, the differences in the tensile fractured surfaces suggest fillers played a role in changing the bio-polymer matrix.

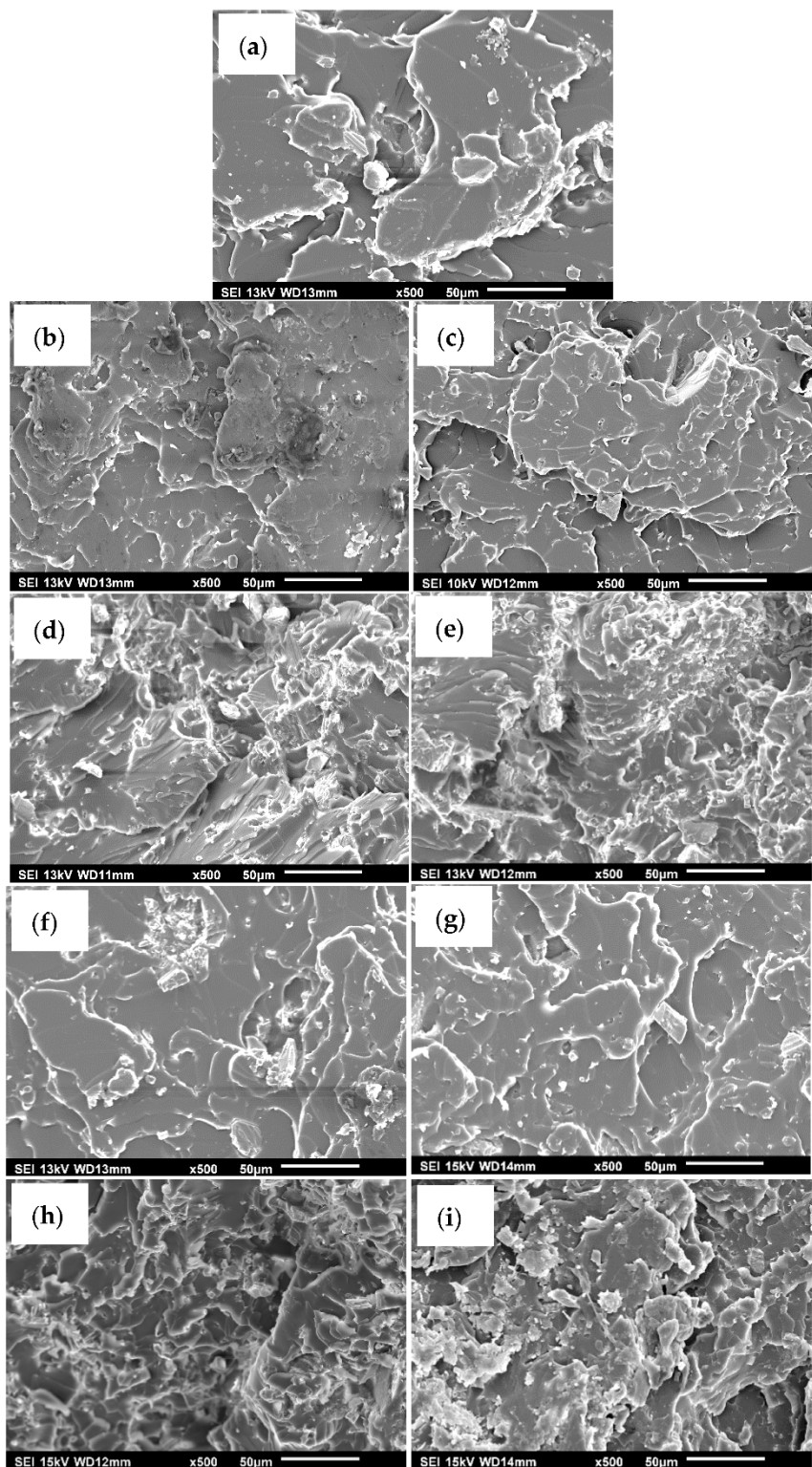

**Figure 4.** SEM micrographs of tensile fractured surfaces of (**a**) unfilled bio-epoxy, (**b**) 5 wt.% ES, (**c**) 5 wt.% ES/SA, (**d**) 20 wt.% ES, (**e**) 20 wt.% ES/SA, (**f**) 5 wt.% LS, (**g**) 5 wt.% LS/SA, (**h**) 20 wt.% LS and (**i**) 20 wt.% LS/SA.

Figure 5 shows the flexural fractured surfaces of the unfilled bio-epoxy (Figure 5a), bio-epoxy filled with 5 wt.% untreated eggshells (Figure 5b), bio-epoxy filled with 5 wt.% eggshells treated with stearic acid (Figure 5c), bio-epoxy filled with 20 wt.% untreated eggshells (Figure 5d), bio-epoxy filled with 20 wt.% eggshells treated with stearic acid

(Figure 5e), bio-epoxy filled with 5 wt.% untreated limestone (Figure 5f), bio-epoxy filled with 5 wt.% limestone treated with stearic acid (Figure 5g), bio-epoxy filled with 20 wt.% untreated limestone (Figure 5h), and bio-epoxy filled with 20 wt.% limestone treated with stearic acid (Figure 5i). The fractured surface of the unfilled bio-epoxy resin presented a smooth, plate-like, and cleavage surface suggesting brittle failure. In a similar manner as the tensile fractured surfaces, the unfilled bio-epoxy presented fewer uninterrupted crack paths after initiation in comparison to the composites. Both untreated and stearic acid treated fillers showed slightly rougher fractured surfaces due to the addition of fillers compared to the unfilled bio-epoxy and further increased with 20 wt.% filler loadings. Greater quantities of filler contents tend to encourage stress concentrations, which generate additional cracks and reduce mechanical strength properties.

### 3.6. Tensile Properties

The effect of untreated eggshell, untreated limestone, eggshell treated stearic acid, and limestone treated stearic acid filler loadings of 5, 10, and 20 wt.% on the tensile strength and tensile modulus of bio-epoxy polymer composites is shown in Figures 6 and 7, respectively. For both Figures 6 and 7, the lower case letters a,b,c, . . . are the results from statistical ranking using Tukey's method at 95% confidence level. The inclusion of eggshell and limestone in the bio-epoxy matrix decreased the tensile strength with an increase in filler loading up to 20 wt.%. In contrast, the tensile modulus tended to increase with filler loadings. The unfilled bio-epoxy had a tensile strength of 60 MPa, which is similar to the manufacturers' data sheet (62 MPa). The untreated fillers with composite loadings of 5, 10, and 20 wt.% reduced in tensile strength by approximately 15%, 25%, and 31%, respectively, for eggshells, whereas the composites with limestone fillers decreased by 16%, 28%, and 32%, respectively, compared to the unfilled bio-epoxy. Interestingly, eggshell composites had slightly better tensile strengths than those of limestone composites. This may be due to the presence of inherent hydroxyl, carboxyl, and amino functional groups contained in the organic eggshell membranes, which act to promote hydrogen bonding with the epoxy matrix [27]. This suggests removal of the organic membranes may not be required because they promote adhesion to the epoxy. Depending on the end use and whether or not the composite is loaded in tension, eggshell or limestone loadings of 5, 10, and 20 wt.% may be acceptable. For instance, assuming the application required a tensile strength of 30 MPa, then all filler loadings would be acceptable such that a higher strength would not be required. The benefit of these composites is to repurpose a waste material while reducing the overall cost of the polymer. However, filler loadings below 5 wt.% would be a better choice. The results showed loadings above 5 wt.% further reduced the composite tensile strength below that of the bio-epoxy matrix. Although the particles were thoroughly mixed into the resin during composite processing, particulate fillers added in greater amounts (e.g., 10 and 20 wt.%) may tend to agglomerate due to electrostatic forces existing between the small particles [28]. Agglomeration suggests poor dispersion of fillers in the matrix. These agglomerates are sites of stress concentration, which aid in crack initiation and propagation to induce brittle failure [12,29]. The results are in agreement with a study that reported a decrease in tensile strengths with a increase in untreated eggshell loadings (particle size of 90 μm) in a polypropylene matrix. The authors believed the reductions were due to stress concentrations as a result of higher filler loadings (e.g., 20, 30, and 40 wt.%) [10]. In contrast, another study reported improved tensile strengths when smaller particle sizes (0.2 μm) of eggshell fillers were added to a polyvinyl chloride matrix at 10 wt.% filler content [30]. The increase in the composite properties may be attributed to the smaller particle size fillers and processing technique [11,31].

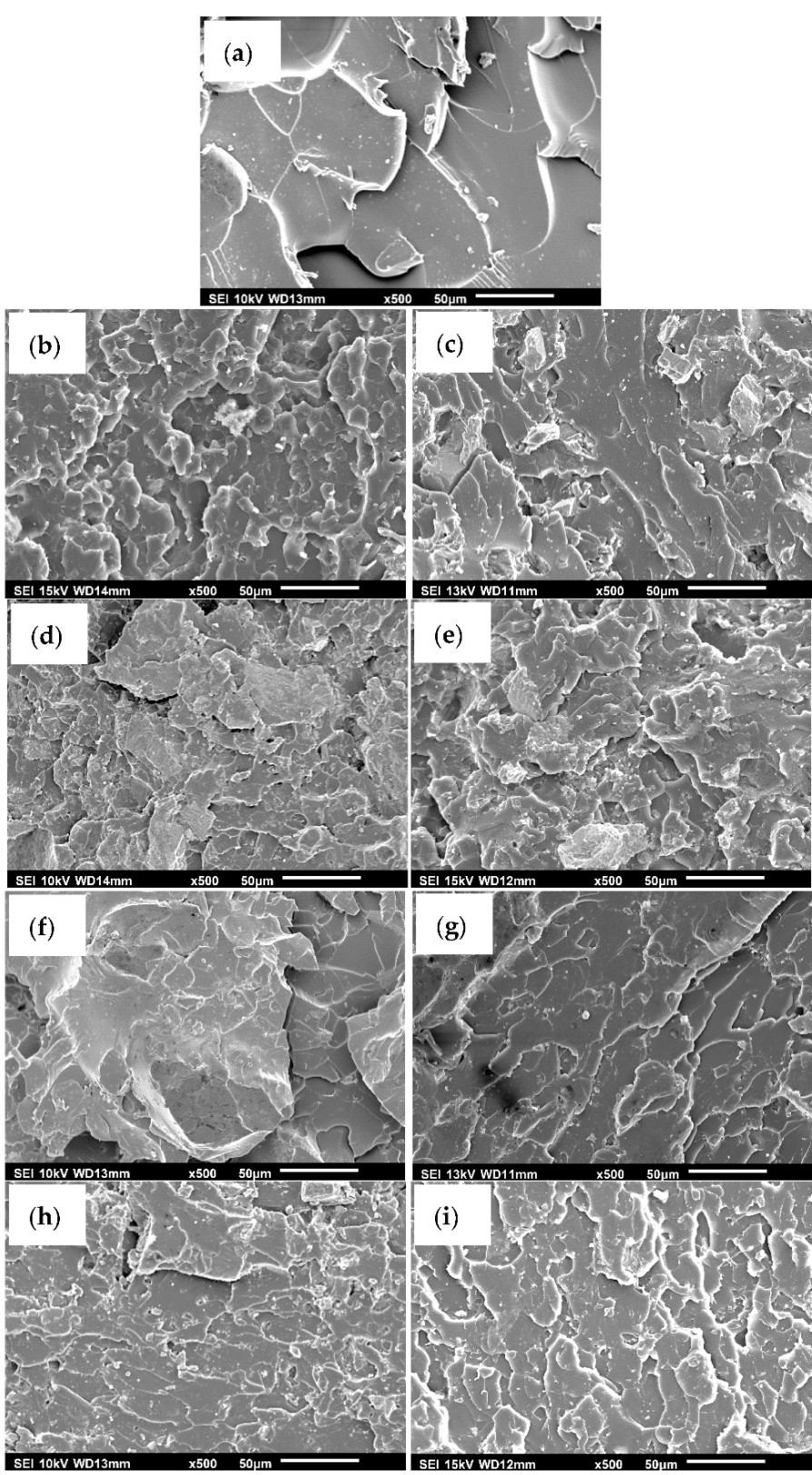

**Figure 5.** SEM micrographs of flexural fractured surfaces of (**a**) unfilled bio-epoxy, (**b**) 5 wt.% ES, (**c**) 5 wt.% ES/SA, (**d**) 20 wt.% ES, (**e**) 20 wt.% ES/SA, (**f**) 5 wt.% LS, (**g**) 5 wt.% LS/SA, (**h**) 20 wt.% LS, and (**i**) 20 wt.% LS/SA.

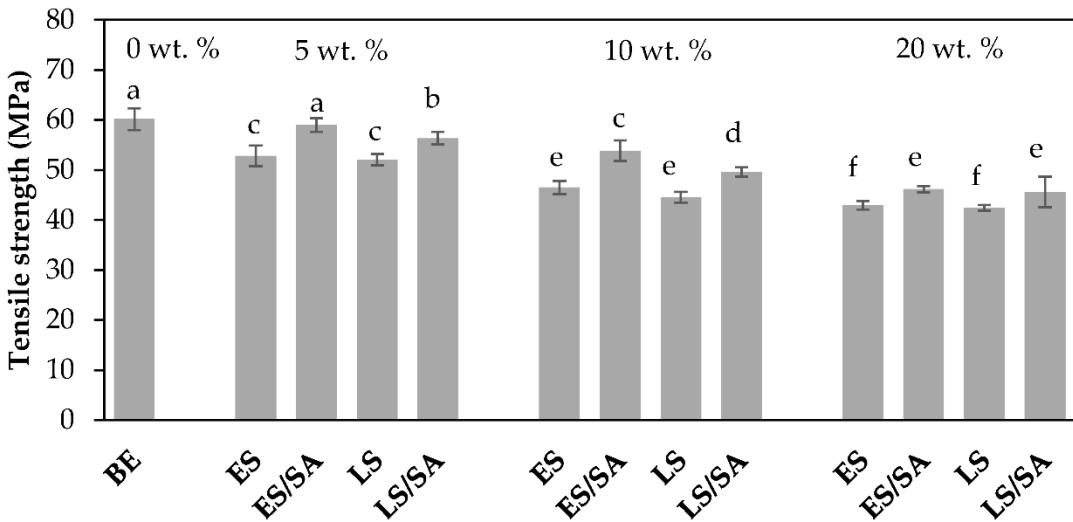

**Figure 6.** Tensile strength of unfilled bio-epoxy and bio-epoxy composites with loadings of 5–20 wt.%, where a,b,c . . . are the results from statistical ranking using Tukey's method at 95% confidence level.

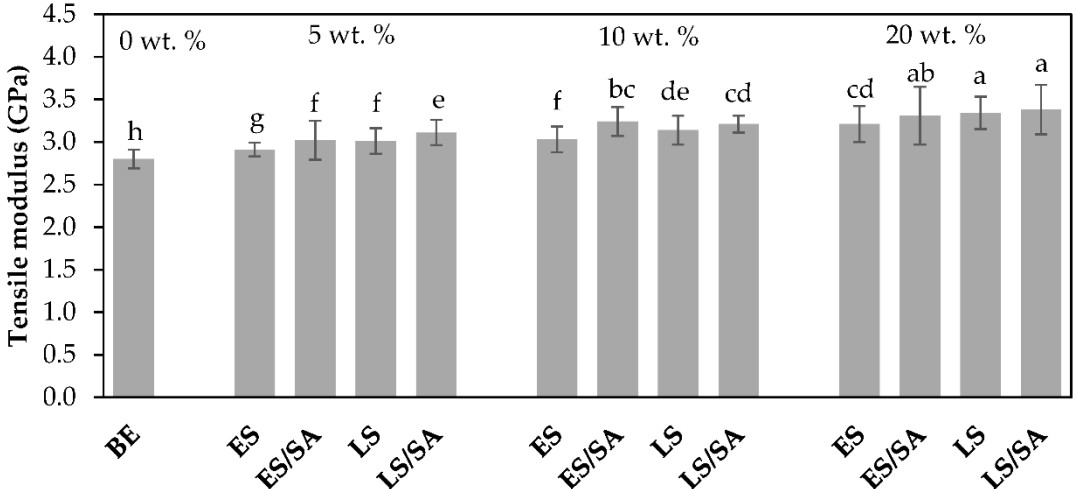

**Figure 7.** Tensile modulus of unfilled bio-epoxy (BE) and bio-epoxy composites with loadings of 5–20 wt.%, where a,b,c . . . are the results from statistical ranking using Tukey's method at 95% confidence level.

In an effort to improve the tensile strengths and reduce agglomeration of $CaCO_3$ particles, the effect of stearic acid treated fillers on the bio-epoxy matrix composites was evaluated. The stearic acid treated fillers with loadings of 5, 10, and 20 wt.% reduced tensile strengths by approximately 5%, 13%, and 26%, respectively, for eggshell fillers, and by 9%, 20%, and 26%, respectively, for limestone fillers compared to the unfilled bio-epoxy. However, the tensile strengths slightly improved by 12%, 16%, and 7% for eggshells treated with stearic acid, and by 5%, 11%, and 7%, respectively, for limestone treated with stearic acid compared to the untreated filler composites. Despite the improvement in the tensile strengths of the stearic acid treated fillers, the composite strength values were not higher than those of the unfilled bio-epoxy matrix. The results suggest the stearic acid surface treatment had a minor effect for improving dispersion of fillers, reducing agglomeration of particles, and promoting a more effective stress transfer from matrix to filler [11]. In a similar work, polypropylene eggshell composites containing average particles of 90 μm with stearic acid surface coatings of 6 wt.% showed an improvement in composite tensile strengths when 10 wt.% filler contents were added compared to the untreated eggshell fillers. The results also showed that higher filler loadings (e.g., 20, 30, and 40 wt.%) tended to further reduce the tensile strengths for both types of fillers (untreated and treated) [10].

Typically, the tensile strength of a particle filled polymer decreases due to a reduce load bearing cross-section from the fillers and by the interaction of the polymer-filler [32]. Filler materials are not always added to polymers to improve the tensile strength but often to reduce the total cost of the more expensive polymer without greatly affecting its properties. A study showed a high-density polyethylene (HDPE) composite with filler size of 65 mesh (or <212 µm) containing 20 wt.% stearic acid-coated eggshells had a moderate tensile strength improvement of about 25% compared to the untreated eggshells. In general, composites containing eggshells treated with stearic acid had better tensile strengths than the composites containing calcium carbonate treated with stearic acid. The authors attributed the results to better interfacial adhesion between the filler and the HDPE matrix for stearic acid-coated fillers [33]. In the current study, the trend was similar, such that the stearic acid treated filler composites had a slightly higher tensile strength than the untreated filler composites. This may be attributed to the adhesion between the bio-epoxy matrix and filler particles. Statistical analysis and ranking of the results for tensile strength shown in alphabetical order in the columns of Figure 6 indicated that the bio-epoxy and the composite with 5 wt.% ES/SA had the highest strength amongst all formulations. The weakest materials in terms of tensile strength were composites containing 20 wt.% ES or LS according to the statistical analysis.

The unfilled bio-epoxy had a tensile modulus of 2.80 GPa, comparable to the manufacturers' data sheet (2.7–3.2 GPa). For the untreated fillers, with loadings of 5, 10, and 20 wt.%, the tensile modulus was enhanced by 4%, 8%, and 15%, respectively, for eggshells and improved by 8%, 12%, and 19%, respectively, for limestone composites compared to the unfilled bio-epoxy. With the addition of a stiffer filler material such as eggshell or limestone, for which the modulus of their mineral $CaCO_3$ has been reported to be 88 GPa [34] compared to 2.80 GPa for pure bio-epoxy, the composites tended to slightly increase in stiffness. The composite tensile modulus followed with an increase in filler loadings for both eggshell and limestone. Similarly, the tensile modulus improved by 8%, 16%, and 18% for eggshells treated with stearic acid, and by 11%, 15%, and 21%, respectively, for limestone treated with stearic acid compared to the untreated filler composites. A related study observed an improvement in tensile modulus for polypropylene composites combined with either untreated eggshells or eggshells treated with stearic acid fillers, where the stearic acid treated filler composites showed the best improvements in tensile modulus [10]. Similar to stearic acid-coated eggshells, a study showed low-density polyethylene (LDPE) composites containing 5 to 25 wt.% of 63 µm size eggshell powders treated with NaOH increased in tensile modulus with the increase in filler loading for both untreated and NaOH treated eggshell fillers compared to the pure LDPE matrix [35]. In a related work, polyethylene composites containing 5–40 wt.% of 25 µm size filler particle loadings exhibited higher tensile modulus compared to pure polyethylene. The influence of eggshells treated with silane and titanate coatings marginally improved the modulus over the uncoated fillers [5]. According to the statistical analysis shown in alphabetical order in the columns of Figure 7, the bio-epoxy had the lowest modulus amongst all composites, which was significantly meaningful (95% confidence level). By comparison, composites with 20 wt.% LS, ES/SA, and LS/SA resulted in the highest tensile modulus, which was statistically meaningful and higher than other formulations.

### 3.7. Flexural Properties

The flexural properties displayed a similar trend to the tensile results. The effect of untreated eggshells, untreated limestone, eggshell treated with stearic acid, and limestone treated with stearic acid filler loadings of 5, 10, and 20 wt.% on the flexural strength and flexural modulus properties of bio-epoxy polymer composites are shown in Figures 8 and 9, respectively. For both Figures 8 and 9, the lower case letters a,b,c, . . . are the results from statistical ranking using Tukey's method at 95% confidence level. The addition of eggshell and limestone to the bio-epoxy matrix tended to decrease the flexural strength and improve the flexural modulus. The bio-epoxy without fillers had the highest flexural

strength of 95.7 MPa, comparable with the manufacturers' data sheet (92.7 MPa). The untreated eggshell and limestone filler types showed similar flexural behaviors. For instance, composites containing 5, 10, and 20 wt.% of untreated fillers decreased by 10%, 25%, and 40%, and by 11%, 24%, and 38% for eggshell and limestone fillers, respectively, compared to the unfilled bio-epoxy. With a 5 wt.% filler content, a drop in strength of 10–11% was observed compared to the unfilled bio-epoxy matrix, signifying larger filler contents are ineffective at improving the flexural strength. Similar to the tensile strength results, clustering of filler particles, which is more significant at higher loadings, may create stress concentration zones causing cracks to develop within the polymer matrix, thus inducing early failure [12,29]. The results are consistent with a previous study that reported an increase in flexural strength with low eggshell loadings (1 and 2 wt.%) of particle sizes less than 10 nm (0.01 µm) in a polyester matrix, whereas at higher loadings (3 wt.%) the strengths reduced [36]. In contrast, a similar study reported a slight improvement (8–10%) in flexural strengths when 5 and 10 wt.% nano-eggshell (specific nanoparticle size was not mentioned) loadings were added to a Super Sap epoxy [25]. Although the eggshell treated with stearic acid and limestone treated with stearic acid filler composites fell slightly below the flexural strength of the unfilled bio-epoxy, the addition of these stearic acid-treated fillers presented a minor increase in flexural strength compared to that of the untreated filler composites. For example, as the eggshell treated with stearic acid filler loadings increased by 5, 10, and 20 wt.%, the composite flexural strengths reduced by approximately 5%, 20%, and 35%, respectively, whereas the limestone-treated stearic acid fillers decreased the composite strengths by 6%, 20%, and 37%, respectively. The stearic acid-treated filler composites containing 5 wt.% loadings presented a reduction of 5–6% (compared to 10–11% for untreated fillers) in flexural strength. This moderate improvement suggests the stearic acid treatment reduced the surface energy of the $CaCO_3$ filler particles, enhanced particle dispersion in the matrix, and promoted good interfacial interaction between the filler and matrix. A study also showed eggshells treated with stearic acid and added to HDPE composites containing 20 wt.% fillers (<212 µm particle size) had better flexural strength than the untreated eggshells. Analogous to the tensile strength results, the composites containing eggshell treated with stearic acid had enhanced flexural strengths compared to the composites with calcium carbonate-treated stearic acid. The authors believed the improvement in flexural strengths may have been due to the increase in the crystallinity of treated HDPE/eggshell composites compared to the untreated ones [33]. In the current study, the XRD results shown in Figure 1 for stearic acid coated fillers do not provide conclusive evidence for a change of crystallinity in the diffraction peak intensities. Statistical analysis for flexural strength indicated that the difference between the bio-epoxy and the composites was significant with higher strength for the bio-epoxy. At each loading level, composites containing ES/SA or LS/SA resulted in superior strength compared to those with ES or LS. Composites with higher percentages of fillers had the lowest flexural strength amongst all, as shown by the statistical analysis at the 95% confidence level.

The flexural modulus of both untreated and stearic acid-treated filler composites improved with the increase in filler loadings compared to the unfilled bio-epoxy, as shown in Figure 9. The flexural modulus of the unfilled bio-epoxy was 2.5 GPa, compared to 2.8 GPa from the manufacturers' data sheet. As the untreated filler loadings increased from 5 to 10 and 20 wt.%, the eggshell composite flexural modulus increased by 8%, 15%, and 15%, respectively, and the limestone composites improved by 11%, 17%, and 19%, respectively. At 20 wt.% filler loadings for both eggshell and limestone composites, the flexural modulus increased by 15 and 19%, respectively, compared to the unfilled bio-epoxy. This improvement may be attributed to the higher stiffness of the limestone filler material. The increase in flexural modulus with eggshell fillers agrees with results obtained by other studies which saw an improvement in a Super Sap epoxy composite with up to 4 wt.% loading [25] and green polyethylene with maximum loadings of 40 wt.% [5]. In the same manner, another study reported an increase in flexural modulus of a GreenPoxy composite with inclusions of ground seashell (mainly composed of calcium carbonate)

fillers up to 30 wt.% loadings [37]. Coating the eggshell and limestone fillers with stearic acid also increased the flexural modulus of the composites slightly above the untreated filler composites. For instance, as the stearic treated filler loadings increased from 5 to 10 and 20 wt.%, the flexural modulus of the eggshell composites improved by 17%, 19%, and 21%, respectively, and the limestone composites increased by 16%, 18%, and 20%, respectively. At 20 wt.% filler loadings, the flexural modulus for both eggshells treated with stearic acid and limestone treated with stearic acid improved by 20% and 21% (15% and 19% for untreated fillers), respectively, compared to the unfilled bio-epoxy. In addition to the rigid $CaCO_3$ particulates, the stearic acid acts to improve the interface between the fillers and the bio-epoxy matrix. A study on stearic acid-coated calcium carbonate particulate composites with sizes in the range of 0.18–0.25 μm were observed to improve in flexural modulus by as much as 30% at a 15 wt.% loading compared to the neat polypropylene matrix [38]. According to the statistical analysis, the flexural modulus of the bio-epoxy was significantly lower than the other composites followed by ES- and LS-filled composites at 5 wt.% loading. At higher filler loadings, the modulus appeared to level off, as shown in Figure 9.

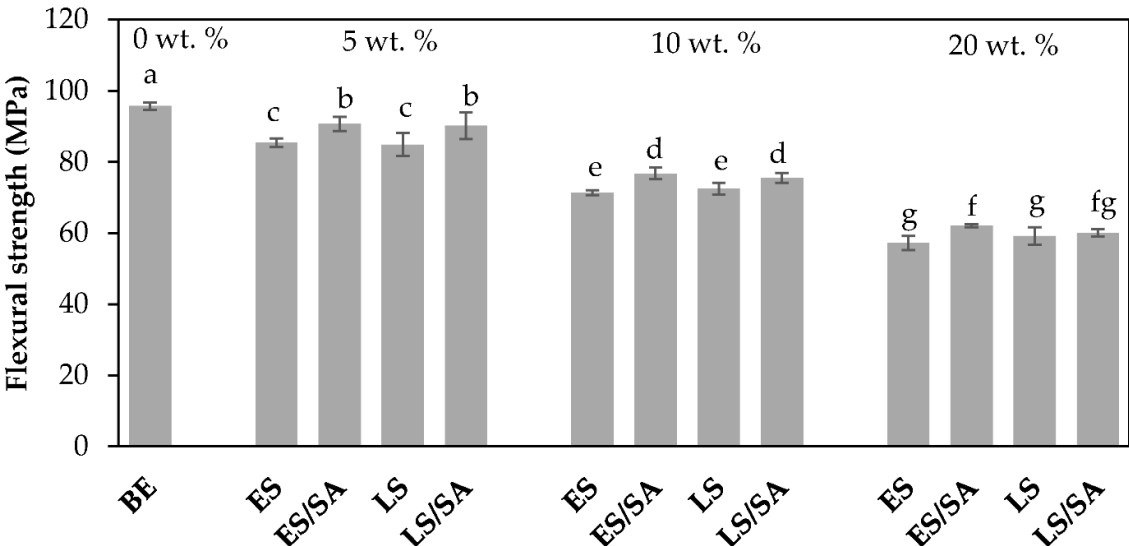

**Figure 8.** Flexural strength of unfilled bio-epoxy and bio-epoxy composites with loadings of 5–20 wt.%, where a,b,c . . . are the results from statistical ranking using Tukey's method at 95% confidence level.

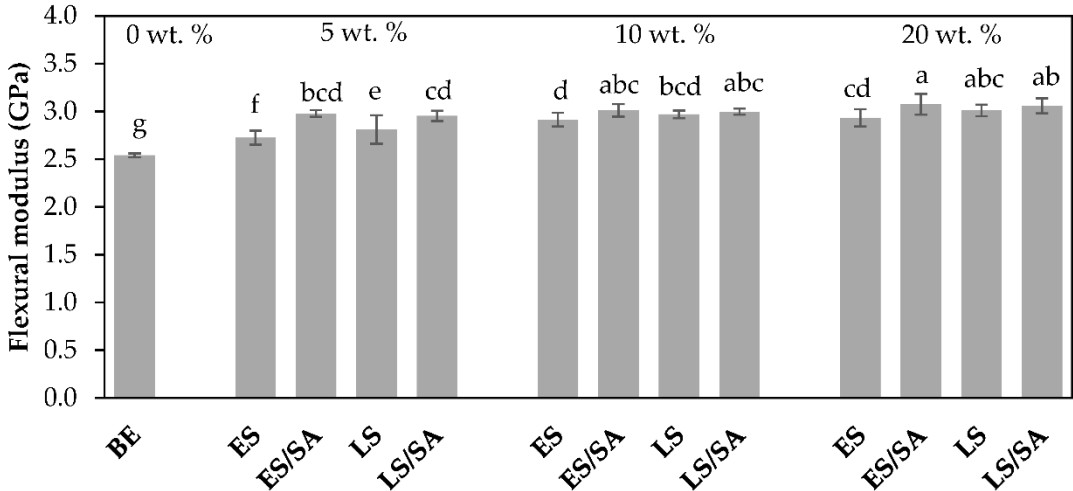

**Figure 9.** Flexural modulus of unfilled bio-epoxy and bio-epoxy composites with loadings of 5–20 wt.%, where a,b,c . . . are the results from statistical ranking using Tukey's method at 95% confidence level.

### 3.8. Charpy Impact Energy Properties

The Charpy test was conducted at room temperature (23 °C) and −40 °C on bio-epoxy composites containing untreated eggshell, untreated limestone, eggshell treated with stearic acid, and limestone treated with stearic acid fillers in amounts of 5, 10, and 20 wt.%. The results are shown in Figures 10 and 11, respectively. For both Figures 10 and 11, the lower case letters a,b,c, . . . are the results from statistical ranking using Tukey's method at 95% confidence level. As shown in Figure 10, at 23 °C, the unfilled bio-epoxy had an impact energy of 8.21 kJ/m$^2$. The energy absorbed by the composites decreased with an increase in filler content for both untreated filler types at 23 °C and −40 °C. As the filler loadings increased from 5 to 10 and 20 wt.%, the Charpy impact energy for composites containing eggshells decreased by 14%, 34%, and 44%, respectively, and the limestone fillers reduced the composite impact toughness by 9%, 31%, and 41%, respectively, compared to the unfilled bio-epoxy. In a similar manner, composites encompassing stearic acid treated fillers also exhibited a decrease in Charpy impact toughness with an increase in filler loadings. For example, as the filler loadings increased from 5 to 10 and 20 wt.%, the Charpy impact toughness of the composites with eggshells treated with stearic acid decreased by 24%, 37%, and 47%, respectively, and that of composites with limestone treated with stearic acid fillers reduced by 20%, 36%, and 43%, respectively, compared to the unfilled bio-epoxy. Similar impact behaviours were reported for a GreenPoxy composite containing untreated seashell filler levels of 5, 10, 20, 30, and 40 wt.%, where a remarkable decrease between 86–92% in toughness was reported compared to the polymer without fillers [37]. Although a study reported excellent particle size reduction and dispersion for polypropylene/eggshell composites, a decrease in impact strength with increasing eggshell content was witnessed. The authors suggested the particles in the polymer matrix intensify the chances for the development and propagation of micro-cracks throughout the matrix during the impact test, which results in the disruption of the polymer-filler interface [39]. The untreated eggshell and untreated limestone composites behaved slightly better than the stearic acid-treated filler composites. A similar study also found the impact strengths were not improved for both eggshells treated with stearic acid and calcium carbonate treated with stearic acid fillers, compared to the untreated composites [33].

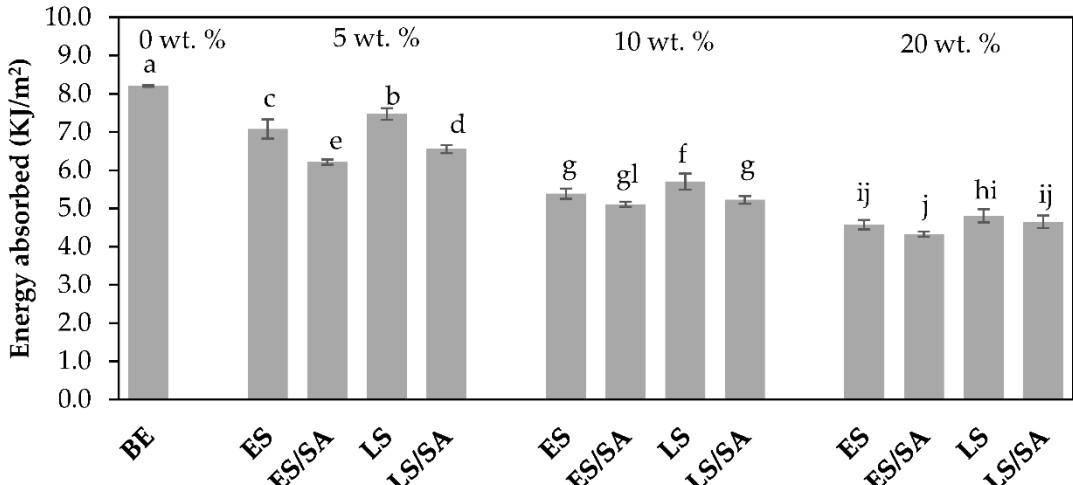

**Figure 10.** Effect of filler loadings on the Charpy impact energy absorbed by unfilled bio-epoxy and bio-epoxy composites with loadings of 5–20 wt.% at 23 °C, where a,b,c . . . are the results from statistical ranking using Tukey's method at 95% confidence level.

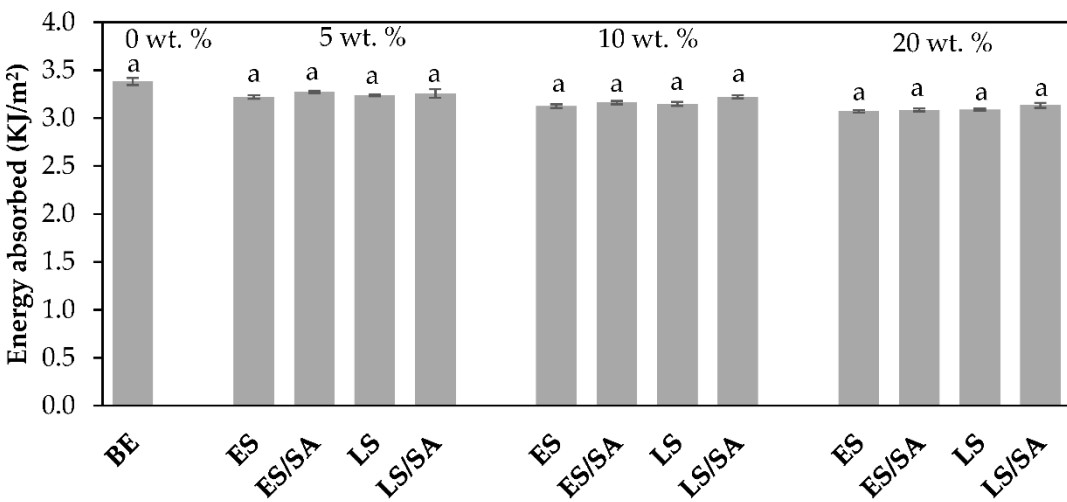

**Figure 11.** Effect of filler loadings on the Charpy impact energy absorbed by unfilled bio-epoxy and bio-epoxy composites with loadings of 5–20 wt.% at −40 °C, where a,b,c ... are the results from statistical ranking using Tukey's method at 95% confidence level.

The impact toughness values decreased remarkably when tested at −40 °C, as shown in Figure 11. For instance, the unfilled bio-epoxy had an impact energy of 3.38 kJ/m$^2$, a 58% decrease from its room temperature value. This suggests the bio-epoxy has an increasingly brittle behavior and is not able to absorb energy before fracture at relatively cold temperatures. With the addition of 5, 10, and 20 wt.% filler loadings, the Charpy impact toughness of the composites did not improve at all filler levels and ranged in values of 3.07–3.24 kJ/m$^2$ for untreated eggshell and untreated limestone filler, and 3.09–3.27 kJ/m$^2$ for stearic acid-treated fillers. For example, the Charpy impact energy of the untreated eggshell composites decreased by 5%, 7%, and 9%, respectively, and that of the untreated limestone composites reduced by 4%, 7%, and 9%, respectively. In a similar manner, both stearic acid-treated fillers also did not positively affect the Charpy impact toughness at any of the filler loadings. For instance, the impact energy in the eggshell treated with stearic acid composites decreased by 3%, 7%, and 9%, respectively, and the energy in limestone treated with stearic acid composites reduced by 4%, 5%, and 7%, respectively. These drops in impact energy agree with a previous study which reported a decrease in impact fracture toughness of an epoxy composite evaluated at a low temperature of −20 °C [40]. According to the DSC results, the T$_g$ of the unfilled bio-epoxy was 56.3 °C (Table 2); −40 °C is well below this value, which suggests the bio-epoxy became less pliable, harder, and more brittle. The results highlight these bio-epoxy composites should not be used in cold weather if the application will be exposed to impact load. However, room temperature applications are more feasible at low filler loadings because the impact energy only decreased by 9–14% for 5 wt.% contents. In terms of statistical analysis, the difference in impact energy (at 23 °C) between composites was significant (in most cases) with lower values for the composites consisting of higher amounts of filler. However, no statistical difference was observed at −40 °C between the control and all composite formulations, as shown in Figures 10 and 11.

**Table 2.** The T$_g$-onset, T$_g$-midpoint and T$_g$-end temperature for pure bio-epoxy and composites with 5 wt.% fillers.

| Composite Formulation | T$_g$-Onset Temperature (°C) | T$_g$-Midpoint Temperature (°C) | T$_g$-End Temperature (°C) |
|---|---|---|---|
| BE | 54.1 | 56.3 | 58.4 |
| 5 wt.% ES | 55.6 | 57.9 | 60.2 |
| 5 wt.% ES/SA | 54.9 | 56.7 | 58.5 |
| 5 wt.% LS | 55.1 | 56.4 | 57.7 |
| 5 wt.% LS/SA | 54.8 | 56.4 | 58.0 |

### 3.9. Differential Scanning Calorimetry (DSC)

DSC was used to evaluate the glass transition temperature ($T_g$) of the pure bio-epoxy and to determine if any deviations occurred due to the addition of the fillers. In this study, unfilled bio-epoxy resin and 5 wt.% filled composites (ES, LS, ES treated with SA, and LS treated with SA) were evaluated for their $T_g$ because they presented better tensile strength, flexural strength, and Charpy impact energy (most promising material) than those with increased filler loadings. The $T_g$-onset, $T_g$-midpoint, and $T_g$-end point temperatures of the unfilled bio-epoxy resin and 5 wt.% composite formulations are given in Table 2. The $T_g$-midpoint temperatures were calculated as an average of the onset and end transition temperatures obtained from the heat flow curves given in Figure 12. The results showed the $T_g$-midpoint obtained for the unfilled bio-epoxy was 56.3 °C, which is within the value reported in the manufacturers' data sheet (54.0 °C). The inclusion of a low concentration of untreated eggshell and limestone fillers into the bio-epoxy matrix had a minor effect on the polymer network, which slightly increased the $T_g$-midpoint compared to unfilled bio-epoxy. For example, the inclusion of 5 wt.% untreated eggshell filler increased the $T_g$-midpoint by 1.6 °C, whereas the 5 wt.% limestone composite, 5 wt.% eggshell treated with stearic acid composite, and the 5 wt.% limestone treated with stearic acid composite saw no significant change in the $T_g$-midpoint in comparison to the unfilled bio-epoxy resin. A study reported similar observations in which there were no significant changes in the glass transition temperatures for synthetic epoxy composites containing 2, 4, and 6 wt.% calcium carbonate fillers [41]. The results suggest that at 5 wt.% there may be not enough filler material distributed within the bio-epoxy matrix to make a substantial change in the $T_g$-midpoint.

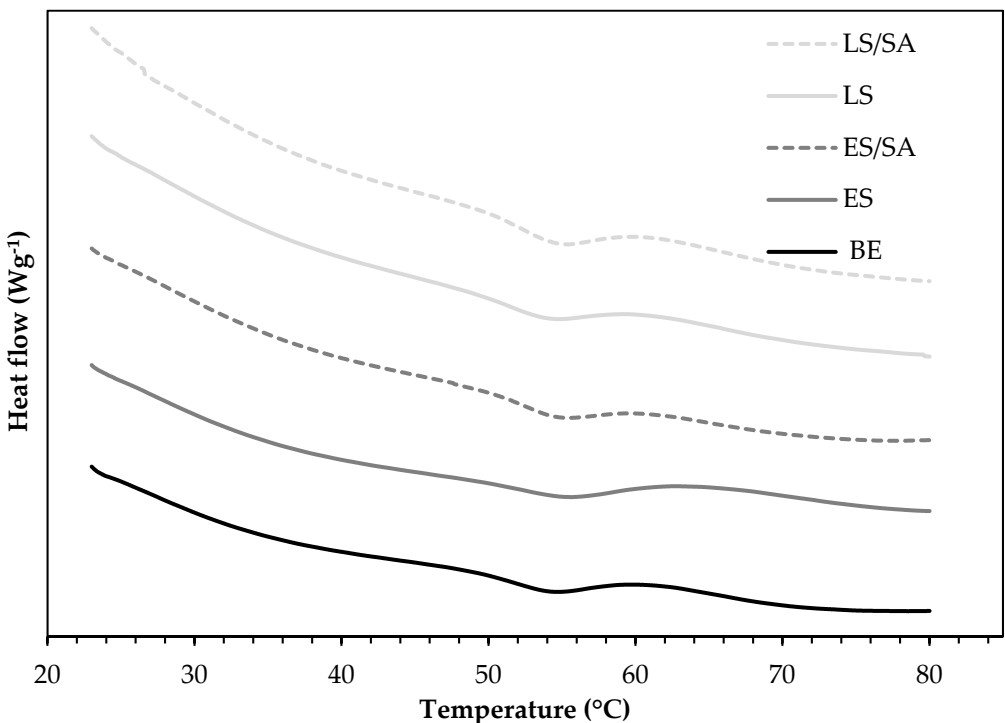

**Figure 12.** DSC thermographs showing the effect of 5 wt.% ES, LS, and SA-treated ES and LS fillers on the $T_g$ of bio-epoxy composites.

The addition of $CaCO_3$ particulate fillers to the bio-epoxy polymer negatively influenced the tensile strength, flexural strength, and impact strength, but had a positive effect on the tensile and flexural modulus. The reduction was generally due to the presence of geometric discontinuities of the filler particles in an otherwise continuous matrix, which acted as stress concentrations that led to the composite failure. In contrast, the composite

tensile and flexural stiffness tended to increase due to the rigid particles being stiffer than the matrix. Future work will entail lower filler contents (<5 wt.% fillers), smaller particle sizes, and moisture and/or water sorption experiments.

## 4. Conclusions

This study evaluated the mechanical properties of eggshell and limestone bio-epoxy composites with different filler loadings for potential filler applications in bio-epoxy resin. The results showed that although the fillers improved tensile and flexural moduli, the disadvantage of adding particulate fillers to polymers is generally a decrease in tensile strength, flexural strength, and Charpy impact strength. The composites containing stearic acid-coated fillers did not have better mechanical properties than the unfilled bio-epoxy, possibly due to the low level of surface coating. The Charpy impact tests were conducted at 23 °C and −40 °C, where the results revealed that cold weather applications in which impact toughness is required are not recommended for this bio-epoxy composite material. Based on the observations in this study, future work can entail lower than 5 wt.% filler contents, because it was shown that at 10 and 20 wt.% loadings, the tensile strength, flexural strength, and Charpy impact degraded quickly. It would also be beneficial to verify if trace elements found in eggshells have an effect on the polymer composite properties.

**Author Contributions:** Conceptualization, D.E.C. and S.O.; methodology, D.E.C.; formal analysis, S.O. and M.S.; investigation, S.O.; resources, D.E.C.; data curation, S.O. and D.E.C.; writing—original draft preparation, D.E.C.; writing—review and editing, D.E.C.; supervision, D.E.C. and M.S.; project administration and funding acquisition, D.E.C. All authors have read and agreed to the published version of the manuscript.

**Funding:** This research was funded by Natural Sciences and Engineering Research Council of Canada (NSERC), Discovery Grant (RGPIN-2020-06701) and the University of Saskatchewan Undergraduate Student Research Assistantship. The APC was funded by MDPI.

**Institutional Review Board Statement:** Not applicable.

**Informed Consent Statement:** Not applicable.

**Data Availability Statement:** The data presented in this study are available on request from the corresponding author. The data are not publicly available due to the raw/processed data required to reproduce these findings cannot be shared at this time as the data also forms part of an ongoing study.

**Conflicts of Interest:** The authors declare no conflict of interest. The funders had no role in the design of the study; in the collection, analyses, or interpretation of data; in the writing of the manuscript, or in the decision to publish the results.

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
