# Peer review of "Fabrication and Characterization of Bio-Epoxy Eggshell Composites"

_2673-3161, doi:10.3390/applmech2040040_

Round 1
Reviewer 1 Report
Please find the as-attached report.

Reviewer 2 Report
The article is interesting because it deals with a relevant topic related to the development of a useful application for waste, namely, eggshell.
- In section 2.2 it is mentioned that “manually agitated and a precipitation process was used to remove additional membranes followed by a drying step” what precipitation method is used?
- In section 3.1, it would be interesting to discuss in depth the problem of removing the eggshell membrane. This is a big problem for industrial applications. Calcination at 450 ºC transforms the particles into a grey-black color due to the carbon that remains in the material. Only at a higher temperature will be possible to eliminate that carbon from the surface. However, at high temperatures, CaCO3 will be transformed into CaO.
- In section 3.1 it is mentioned that 88 wt.% is CaCO3, but in the Introduction it is mentioned in line 47 that “waste chicken eggshells showed they consisted of 94-96 wt. % CaCO3“ Give a comment on that.
- All the results reported in sections “3.5 -Tensile Properties” to “3.7. Charpy Impact Energy Properties” need a statistical analysis to determine which results are equal and different. For example, in Figure 5 Tensile strength of unfilled bio-epoxy and bio-epoxy composites with loadings of 5-20 wt. %., the results for BE are better or equal to the other cases (ES, LS, ES/SA and LS/SA). For example, use the Tukey method can be used to find means that are significantly different from each other. Equal means should be marked with the same letter, while different means marked with different letters. After that, go through the manuscript and see what statements require to be changed accordingly. The method must be applied to Fig. 5 to Fig.10 and maybe also to results in Table 2.
Reviewer 3 Report
- There are similarities in a published thesis from same author especially in the introduction part. The thesis should at least be cited in this manuscript and the introduction preferably revised.
- Experimental part at line 118 should be clarified. It is clear that 2 wt % stearic acid was loaded. However, the wordings is ambiguous because of the dilution with ethanol, making it appear that 2% is even much lower in the final material. Specify whether a mother solution is prepared in this case.
- Particle size distributions should be reported, apart from the sieve particles. This could further aid in the discussion and analyses of results.
- Line 283 should be broken down in component parts for clarity
- Line 402. The trend is inconclusive as only 5% was reported as the low filler region. If this meant as the filler is decreased, the 10 and 20% is just with in experimental errors.
- In line 410, the unfilled BE appears to be still the best choice. What will be the benefit if otherwise?
- Line 452 is not supported by experimental data. Authors are advised to look on the matrix used in this study as one possible cause of the difference with the arguments in the cited reference.
- Data in figures should be rearranged as to show results between ES and ES/SA, LS and LS/SA. This will better reflect the results as the CaCO3 content is not the same on solid basis between ES and LS.
- To further the discussion, can line 541 be corroborated from the X-ray diffraction results of this study?
- The conclusion is to be slightly modified as line 688 is not the focus of this work but of a previous one. The conclusion is not supported by experiment as only 2% was used in this work.
Reviewer 4 Report
Comments
The manuscript number, titled “Fabrication and Characterization of Bio-epoxy Eggshell Composites” discusses the waste material as filler, the effect of coating, and their mechanical properties. The manuscript is well-written and well-presented. I, therefore, recommend to be published after minor revision.
- How do the authors confirm the crystalline phase of CaCO3 is calcite? The authors should provide a significant reason?
- Please refer to the following papers Compos. Struct.1 045004, https://doi.org/10.1016/j.ijhydene.2020.10.139,and https://doi.org/10.1016/j.jallcom.2016.12.360 and accommodate them in the proper place.
- Does the viscosity of SA affect the mechanical properties of the prepared composites?
- There are traces of Magnesium (0.27 %) and Phosphorous (0.13 %). Is there any role of these trace elements while fabricating composites and their mechanical properties?
Round 2
Reviewer 2 Report
The authors addressed the questions raised in the first round. Thus the paper may be published.
Reviewer 3 Report
The manuscript has been sufficiently improved to warrant publication in Applied Mechanics. Minor editing required.
1. Fix caption for Figure 8. Place caption near the intended figure.